# Influence of Technological Stages of Preparation of Rooster Semen for Short-Term and Long-Term Storage on Its Quality Characteristics

Yulia Silyukova * , Elena Fedorova and Olga Stanishevskaya

Russian Research Institute of Farm Animal Genetics and Breeding—Branch of the LK Ernst Federal Research Center for Animal Husbandry, Moskovskoe Shosse, 55a, Pushkin, 196625 St. Petersburg, Russia
* Correspondence: svadim33@mail.ru

**Abstract:** There is a problem of declining quality of rooster semen in the "native semen-equilibrium-short-term and long-term storage (cryopreservation)" cycle. The aim of this study was to determine the effects of various methods of preparing rooster semen on its qualitative characteristics, taking into account the method of removing possible contaminants (centrifugation or filtration), and to evaluate the change in the composition of the cytosol of the spermatozoon of the native semen, during equilibration of the diluted semen and during short-term storage. In this study, semen from roosters ($n$ = 22) of the Russian White breed was used. Experiment 1: semen was divided into 3 aliquots: I—was diluted with synthetic cryoprotective medium (1:1 with LCM control, II—was filtered (membrane pore Ø 0.2 μm), and III—was centrifugated (at 3000 rpm for 10 min). Native and frozen/thawed semen was evaluated. Experiment 2: the composition of carbohydrates and polyols of the spermatozoa of native semen was evaluated during equilibration and after storage (3 h). The results of Experiment 1 showed an advantage in the quality of filtered semen compared to centrifuged in terms of progressive motility (41.0% vs. 27.0%) and chromatin integrity (56.6% vs. 33.6%). Results from frozen/thawed samples of filtered semen compared to centrifuged in terms of progressive motility were 25.5% vs. 5.5%, respectively, and in terms of chromatin integrity—83.5% vs. 64.4%, respectively. The results of Experiment 2 showed the main component in the composition of the native spermatozoa cytosol in assessing the content of carbohydrates and polyols was inositol—75.6%. The content of inositol decreased during storage by 6.5 times (from 0.030 mg/mL to 0.007 mg/mL), proposing the role of inositol as the main antioxidant in the cytosol of spermatozoa, which makes it biologically justified to introduce inositol into the composition of synthetic diluents, including cryoprotective ones.

**Keywords:** spermatozoa; rooster; cytosol; carbohydrates; polyols; inositol; semen storage





## 1. Introduction

For the purposes of artificial insemination, rooster semen is evaluated according to the following parameters: ejaculate volume and sperm concentration, total motility (TM) and progressive motility (PM) of spermatozoa, viability (membrane integrity), chromatin integrity, morphological integrity, and acrosome integrity. Modern researchers often pursue more in-depth studies of the semen, such as the mitochondrial potential of the reproductive cell, the concentration of reactive oxygen species (ROS) [1], the degree of lipid peroxidation (LPO) [2], total antioxidant capacity (TAC) [3], and enzymatic superoxide dismutase (SOD) activity [4].

In the practical, routine process of artificial insemination, the most applicable evaluation characteristics are ejaculate volume and sperm concentration, TM, PM, and sperm viability. They are in high use at the stages of initial evaluation of rooster semen and for artificial insemination, as well as for the subsequent process of its cryopreservation.

The peculiarity of the structure of the reproductive system of roosters and its rudimentary copulatory organ dictates that special attention be paid to the assessment of ejaculate contamination by pollutants of various etiologies. Contaminated semen may contain other bacterial species such as *Staphylococcus* spp., *Coliforms*, and *Streptococcus* spp. [5,6]. Since there is a possibility that the resulting semen may be contaminated with traces of urine and/or droppings, as well as somatic cells (erythrocytes, lymphocytes, epithelial, microbial, and others), the high-quality work of specialists in obtaining semen from roosters is required. Contaminated semen tends to have lower fertility coupled with a high risk of infection in the genital tract, as the polluted/contaminated semen enters the hen's oviduct directly. The negative impact of pollution in the semen of roosters is a decrease in sperm motility, an increase in the number and volume of aggregated spermatozoa conglomerates, a violation of the integrity of cell membranes and DNA due to an increase in the degree of lipid peroxidation, and a high level of ROS, which has been proven by numerous studies [7]. In the conditions of breeding poultry farms, such risks are unacceptable.

There are several ways to eliminate semen contamination. One of them is centrifugation, i.e., complete removal of contaminated seminal plasma with its subsequent replacement with a synthetic medium of full composition. Sanitization of rooster semen with the help of antimicrobials can also be carried out, which in most cases leads to a decrease in the fertilizing ability of the spermatozoa and an increase in the resistance of pathogenic microorganisms to the antibacterial drugs used.

In published materials, researchers disagree on the importance of seminal plasma for both artificial insemination and the cryopreservation of rooster semen. Seminal plasma is a key body fluid that modulates sperm function in all animal species, but the role of seminal plasma in storing avian sperm in vitro remains largely a subject of controversy, since both inhibitory and stimulatory effects of its influence on the efficiency of artificial insemination and both short-term and long-term storage of sperm have been found [8,9]. Recent studies in the development of biochemical and ohmic assessment methods in sperm biology have improved knowledge of the initial components and underlying mechanisms [10,11]. In studies of the composition of the seminal plasma of roosters, the presence of specific proteins ovotransferrin and gallinacin-9, which have an antibacterial function, has been revealed [12,13]. Rooster semen is characterized by aerobic oxidation during ejaculation, so seminal plasma also has a number of protective components that can neutralize excessive ROS production [13]. The composition of seminal plasma includes carbohydrates and polyols as energy and plastic components. It has been established that impaired fertility of rooster spermatozoa may be associated with a change in the content of the carbohydrate components of their glycocalyx [6]. The role of the surface carbohydrates of rooster spermatozoa in maintaining their fertility during short-term and long-term storage is well-covered in the literature [14,15]. However, no less important is the study of the composition of the cytosol of spermatozoa and its changes under the influence of technological factors.

The aim of this study was to determine the effect of various methods of preparing rooster semen on the qualitative characteristics of spermatozoa during short-term storage and cryopreservation, taking into account the method of removing possible contaminants (centrifugation or filtration), and to evaluate the change in the composition of the cytosol of the spermatozoon of native semen after the equilibration of the diluted semen and during short-term storage.

## 2. Materials and Methods

### 2.1. Ethics Endorsement

This study was agreed to and approved by the ethics committee of the Russian Research Institute of Farm Animal Genetics and Breeding—Branch of the L.K. Ernst Federal Research Center for Animal Husbandry (RRIFAGB, Russia) in accordance with the accepted principles of animal bioethics 5 part 2 of the European Convention for the Protection of Vertebrate Animals for Experimental and Other Scientific Purposes (ETS 123 1986).

## 2.2. Animals

The studies were carried out on egg-laying roosters at age 60–64 weeks ($n\male$ = 22) of the Russian White breed in the Centre for Collective Usage "Genetic collection of rare and endangered chicken breeds," RRIFAGB. The roosters were kept in individual cages. Feeding, watering, and light regime corresponded to the age and direction of the bird's productivity. Roosters, starting from the age of puberty, were accustomed to abdominal massage and used in the semen selection regimen 2 times a week. The experiment was carried out in the period May–July 2022.

## 2.3. Design of Technological Stages of the Experiment

**Experiment 1.** Influence of rooster semen preparation methods on the qualitative characteristics of rooster spermatozoa during short-term and long-term storage.

In the experiment, individual indicators of native semen (volume (ml), concentration (billion/mL), and total motility (%)) were evaluated. The obtained and selected ejaculates ($n$ = 22) were combined and divided into 3 aliquots for the following operations: aliquot I—diluted 1:1 with LCM (Leningradskaya Cryoprotective Media) cryoprotective control medium; aliquot II (filtration)—semen filtration was carried out using a reusable filter nozzle SWINNEX® from Millipore (Merck KGaA, Darmstadt, Germania) ) with a membrane pore diameter of 0.2 μm, then diluted 1:1 with LCM cryoprotective control medium; aliquot III (centrifugation) was centrifuged at 3000 rpm for 10 min and the seminal plasma was removed with a graduated pipette, brought to the primary volume with the LCM-control diluent, and again diluted in a ratio of 1:1 with the LCM cryoprotective control medium. All stages of the experiment were performed in 3 repetitions.

**Experiment 2.** The obtained and selected ejaculates ($n$ = 22) were combined and divided into 3 aliquots to assess the composition of carbohydrates and polyols in the cytosol of spermatozoa: aliquot I (native)—native undiluted semen; aliquot II (equilibration)—the semen was diluted 1:1 with LCM cryoprotective control medium and equilibrated for 40 min; aliquot III (storage)—diluted semen was stored for 3 h at 5 °C. The assessment was carried out in 2 repetitions.

## 2.4. Assessment of Semen Quality in the Field

The following criteria were included in the assessment of macroscopic indicators under production conditions: ejaculate volume, spermatozoa concentration, organoleptic visual assessment (color and odor), and total spermatozoa motility (TM). The semen was evaluated in accordance with GOST 27267-2017, "Products for reproduction. Non-diluted fresh sperm of cocks and turkey-cocks. Specifications" [16]. The ejaculate volume was measured with a graduated pipette, the concentration was measured using an Accuread photometer (IMV Technologies, Bellshill, UK, 2019), and the total motility (%) was assessed on a Micromed MC-12 microscope (Micromed®, Ningbo Shengheng Optics & Electronics Co.,Ltd, Gao Qiao Town, Yin County, Ningbo, China) at a magnification of x200, for which 10 μL of each sample, pre-diluted to 1:20, was placed on a heated (37 °C) object glass and covered with a coverslip.

## 2.5. Reagents and Diluent Composition

The reagents used in the study were from Sigma (St. Louis, MO, USA). For dilution of the semen and subsequent freezing, an LCM cryoprotective control medium of the following composition was used. Medium composition per 100 mL of distilled water: monosodium glutamate 1.92 g, fructose 0.8 g, potassium acetate 0.5 g, polyvinylpyrrolidone 0.3 g, protamine sulfate 0.032 g [17].

## 2.6. Sperm Cryopreservation and Thawing

The diluted semen samples were equilibrated from room temperature to 5 °C within 40 min. Dimethylacetamide (DMA) was used as a cryoprotectant in an amount of 6% of the total volume of semen diluted 1:1 with an LCM cryoprotective control medium.

Subsequent incubation was continued for 1 min at 5 °C. Freezing was carried out in granules by separating a drop from the tip of a glass Pasteur pipette. The position of the glass Pasteur pipette with sperm was controlled by a digital hand-held thermometer with a sensor (THERM 2420, AHLBORN, Holzkirchen, Germany) and the temperature in the placement area was −15 to −20 °C, the temperature on the liquid nitrogen surface was −135 °C. The frozen semen was stored for 30 days at −196 °C. The thawing of the granules was carried out on a heated metal plate at 60 °C (device developed by RRIFAGB, 1989).

### 2.7. Quantitative Chromatographic Analysis of Spermatozoa Cytosol Carbohydrates

During the preparation of samples for chromatographic analysis, the aliquots were centrifuged for 10 min at 3000 rpm, the supernatant was removed, and the centrifugate was washed with 0.9% sodium chloride solution and repeated 3 times. The centrifuged semen was then filtered through a nylon filter for 30 min. The prepared centrifuge samples and supernatants were frozen and stored at −25 °C.

The dry weight of the biosample was determined. Extraction of soluble sugars from the spermatozoa (300–400 mg wet weight) was carried out in 5 mL of water at 100 °C for 20 min four times. Proteins were removed from the combined extract [18]. Purification of the carbohydrate extract from charged compounds was carried out on a combined column with ion-exchange resins Dowex-1 (acetate form) and Dowex 50 W (H+). The composition of carbohydrates was determined by gas-liquid chromatography [19]. Merck A-methyl-D-mannoside (Darmstadt, Germany) was used as a standard. Chromatography was carried out on a Kristall 5000.1 gas-liquid chromatograph from CJSC Khromatek (Yoshkar-Ola, Russia) and on a ZB-5 capillary column 30 m, 0.32 mm, 0.25 μm from Phenomenex (Torrance, CA, USA) according to the temperature program from 130 to 270 °C at a speed of 5–6 deg/min. Glycerol, glucose, inositol, fructose, and mannitol (Sigma, St. Louis, MO, USA) were used as markers. The evaluation was carried out twice.

### 2.8. Viability of Spermatozoa of Native and Frozen/Thawed Semen

Spermatozoa viability was analyzed by staining with eosin/nigrosine dye, visualized on an Axio Imager 1.0 phase-contrast microscope (Carl Zeiss Microscopy GmbH, Jena, Germany, 1000× with oil immersion) [20]. At least 200 cells were evaluated in each sample. Cells stained pink were considered damaged (dead), counted as % of the total.

The analysis of total and progressive spermatozoa motility (% of total) was carried out using the CASA ArgusSoft imaging system (Saint-Petersburg, Russia, 2020).

### 2.9. Sperm Chromatin Integrity of Native and Frozen/Thawed Semen

Chromatin integrity was determined using the toluidine blue protocol (TB) [21,22]. Semen smears pre-diluted 1:20 were air-dried, fixed with freshly prepared 96% ethanol-acetone (1:1) at 4 °C for 30 min, and air-dried again at room temperature for 30 min. After that, hydrolysis was carried out with 0.1 M HCl at 4 °C for 5 min and washed three times with distilled water (2 min each time). Samples were stained with 0.05% TB for 20 min at room temperature. The staining buffer consisted of 50% citrate-phosphate buffer (McIlvaine, pH 3.5). Slides were briefly washed with distilled water and lightly blotted with filter paper. At least 200 cells were evaluated in each sample, using a phase contrast optical microscope (Motic BA410E, China, 2019, negative contrast) at x40 magnification. Spermatozoa with intact chromatin were stained light blue, spermatozoa with damaged chromatin were dark blue.

### 2.10. Integrity of Spermatozoa Acrosomes of Native and Frozen/Thawed Semen

Intact sperm acrosomes were determined according to the following protocol: a drop of pre-diluted 1:20 semen was dried in air on glass slides, fixed with 5% paraformaldehyde in phosphate-buffered saline (PBS; pH 7.4) for 15 min, and washed once with PBS. Slides were stained for 5 min with an aqueous solution of 0.25% Coomassie Brilliant Blue R 250 in 10% glacial acetic acid and 25% methanol; they were then washed with distilled water and

covered with coverslips under mounting medium (Lerner Labs Inc., Pittsburgh, PA, USA). Intact spermatozoa acrosomes were stained blue, unstained acrosomes were damaged. At least 200 cells in total were evaluated from 5–6 microscopic fields [23,24].

### 2.11. Degree of Agglutination of Spermatozoa of Native and Frozen/Thawed Semen

The degree of agglutination was determined by the number of aggregated spermatozoa in the conglomerate relative to the total number of spermatozoa in the field of view at x100 magnification using a sperm analyzer (CASA ArgusSoft imaging system, Agussoft Company LLC, St. Petersburg, Russia); at least 5 fields of view were evaluated.

### 2.12. Statistical Analysis

Comparison of viability, motility, and chromatin and acrosome integrity between fresh and frozen/thawed spermatozoa was performed using paired *t*-tests. Differences between the samples were assessed by the Student's method. Results are expressed as mean $\pm$ standard error of the mean (SEM). For statistical analysis, 3 replicates were performed for each trial.

## 3. Results and Discussion

**Experiment 1.** During the formation of the experimental flock, an assessment was made of individual ejaculates of roosters (Table 1). The experiment included roosters with total sperm motility from 61.5% to 94.0% (*CV* 12.1%); according to the assessment of viability, the limits ranged from 44.3% to 86.4% (*CV* 23.2%).

**Table 1.** Indicators of qualitative assessment of individual ejaculates of native semen of Russian White roosters (Maen $\pm$ SEM).

| *n* ♂ | Ejaculate Volume, mL | Concentration, Billion/mL | Total Motility (TM), % | Progressive Motility (PM), % | Viability, % | Chromatin Integrity, % |
|---|---|---|---|---|---|---|
| 22 | $0.38 \pm 0.026$ | $2.08 \pm 0.18$ | $84.3 \pm 2.23$ | $66.5 \pm 2.5$ | $72.9 \pm 5.6$ | $81.1 \pm 1.0$ |

In general, the results showed a high repeatability of the assessment (three replicates); the qualitative characteristics of the semen were little changed for each experimental sample.

Short-term storage of rooster semen.

The variability of semen quality indicators (Table 2) after storage for 3 h at a temperature of 5 °C was aliquot I viability: *CV* 8.8%, PM *CV* 7.8%; aliquot II viability: *CV* 23.9%, *CV* 7.6%; aliquot III viability: *CV* 18.0%, PM *CV* 9.0%.

**Table 2.** Results of semen evaluation after storage for 3 h at 5 °C (Mean $\pm$ SEM).

| Qualitative Indicator | Aliquot I (LCM-Control) | Aliquot II (Filtering) | Aliquot III (Centrifugation) |
|---|---|---|---|
| concentration ppb/ml | $1.82 \pm 0.05$ | $1.71 \pm 0.08$ | $1.23 \pm 0.11$ |
| *CV* concentration, % | 5.3 | 9.4 | 18.3 |
| decrease in concentration, % | 0 | 6.0 | 33.4 |
| total motility, % | $80.0 \pm 1.47$ [a] | $79.8 \pm 0.6$ [a] | $69.5 \pm 0.79$ [b] |
| *CV* total motility, % | 3.7 | 1.5 | 2.3 |
| progressive motility, % | $39.2 \pm 1.52$ [a] | $41.0 \pm 1.56$ [a] | $27.0 \pm 1.21$ [b] |
| *CV* progressive motility, % | 7.8 | 7.6 | 9.0 |
| viability, % | $65.2 \pm 0.07$ | $58.6 \pm 0.08$ | $59.2 \pm 0.03$ |
| *CV* viability, % | 8.8 | 23.9 | 18.0 |
| chromatin integrity, % | $30.9 \pm 4.12$ [a] | $56.6 \pm 0.75$ [b] | $33.6 \pm 2.85$ [a] |
| *CV* chromatin integrity, % | 23.1 | 2.3 | 14.7 |
| acrosome integrity, % | $98.5 \pm 0.26$ | $96.3 \pm 0.49$ | $98.1 \pm 0.34$ |
| *CV* acrosome integrity, % | 0.5 | 0.9 | 0.6 |

Note: a,b *p* < 0.001.

From the data in Table 2, it follows that the qualitative characteristics of the semen during its short-term storage after the removal of contaminants by filtration and centrifugation differ. During centrifugation, the concentration of spermatozoa, progressive mobility, and integrity of chromatin decreased, which was possibly associated with mechanical damage during centrifugation.

However, when storing the LCM-control semen for 3 h at a temperature of 5 °C, a decrease in the integrity of chromatin by 1.8 times was observed, compared with the filtered one. Experiment 2 was devoted to studying the reasons for this fact.

Frozen/thawed semen.

When assessing the integrity of the chromatin of frozen/thawed spermatozoa, a change in the density of chromatin staining in the cell and the intensity of staining was visually noted (Figure 1a,b), depending on the method of semen preparation. Differences in the viability of frozen/thawed versus native semen are shown in Figure 2. The cells of aliquot I (LCM control) and aliquot II (filtration) did not visually differ in density and intensity of chromatin staining.

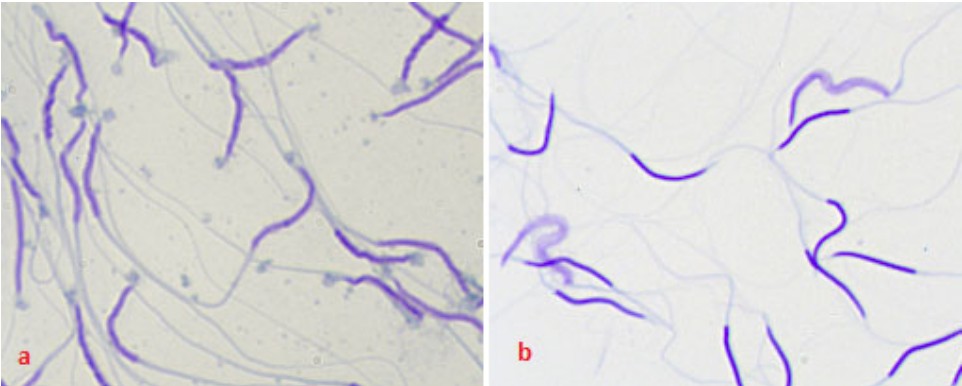

**Figure 1.** Damage to the integrity of chromatin (TBA) of spermatozoa under the influence of technological factors in the freezing protocol (**a**) thawed spermatozoa, LCM control, (**b**) thawed centrifuged spermatozoa. Classification: spermatozoa stained light blue were considered to have normal chromatin and the spermatozoa stained dark blue to violet were considered to have abnormal chromatin.

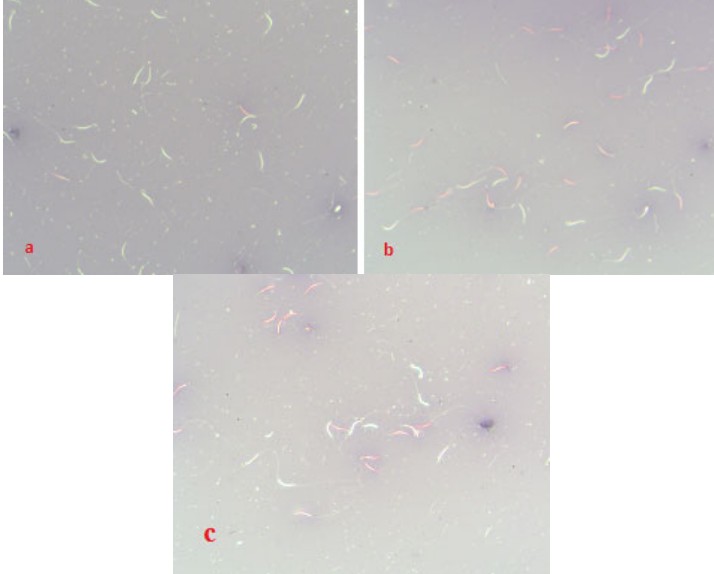

**Figure 2.** Evaluation of viability of spermatozoa (eosin/nigrosin) under the influence of technological factors: (**a**) native diluted semen (filtration); (**b**) frozen/thawed semen (filtration); (**c**) frozen/thawed semen (centrifugation). Classification: pink-stained spermatozoa were considered damaged (dead), white cells were considered intact (alive).

Evaluation of frozen/thawed semen in is presented in Table 3.

**Table 3.** The results of the assessment of frozen/thawed semen under the influence of technological factors (Mean ±SEM).

| Qualitative Indicator | Aliquot I (LCM Control) | Aliquot II (Filtering) | Aliquot III (Centrifugation) |
|---|---|---|---|
| total motility, % | 44.7 ± 5.2 [ab] | 52.2 ± 2.3 [a] | 30.4 ± 3.6 [b] |
| progressive motility, % | 21.4 ± 0.9 [a] | 25.5 ± 2.5 [a] | 5.5 ± 0.4 [b] |
| viability, % | 26.8 ± 0.4 [a] | 40.1 ± 1.7 [b] | 33.4 ± 2.6 [b] |
| chromatin integrity, % | 90.4 ± 6.3 [a] | 83.5 ± 7.5 [a] | 64.4 ± 2.1 [b] |
| acrosome integrity, % | 68.2 ± 1.1 [a] | 68.7 ± 3.1 [a] | 56.9 ± 2.0 [b] |

Note: a, b $p < 0.001$.

The best quality indicators of frozen/thawed semen were in aliquot II (filtered).

The influence of storage regimes on the integrity of acrosomes is shown in Figure 3. The preservation rates of acrosomes in centrifuged frozen/thawed semen confirm the shortcomings of centrifugation as a method of ejaculate purification.

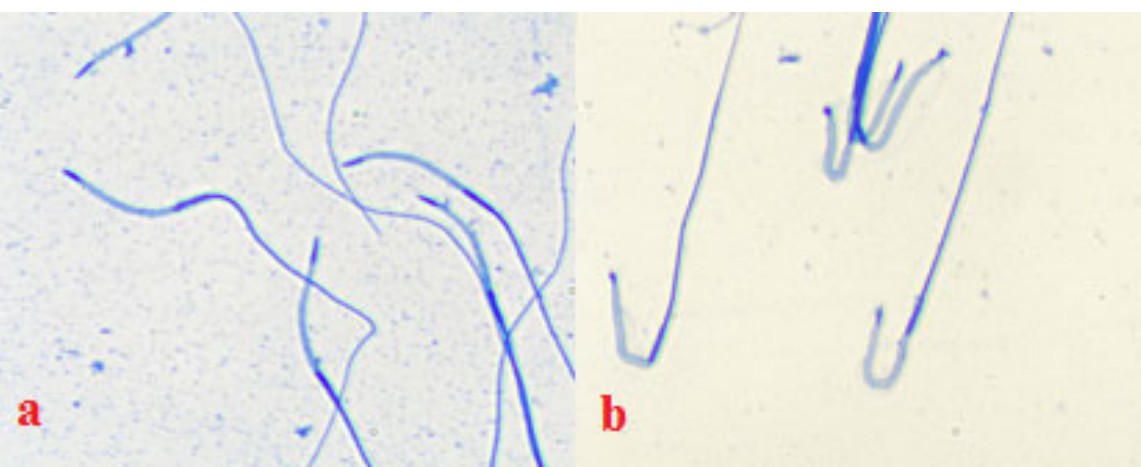

**Figure 3.** Damage to spermatozoa under the influence of technological factors: changes in intactness of the acrosomes of spermatozoa (Coomassie) in the LCM control: (**a**) storage for 3 h (**b**) thawed spermatozoa.

Of all the technological stages, the greatest damage to spermatozoa acrosomes was noted at the stage of freezing/thawing.

The degree of agglutination of frozen/thawed semen, depending on the method of preparation, ranged from 5.0% to 14.0% (Figure 4).

The results of Experiment 1 showed that in semen with seminal plasma removed (centrifugation) during short-term storage, the total and progressive motility of spermatozoa were 10.3% and 14.0% (respectively) lower than that of filtered semen (aliquot II) and 10.6% and 12.2% (respectively) lower than in the LCM control. There was also an overall decrease in general and progressive motility in all groups compared to the indicators of motility in the native semen. The obtained results are consistent with the data of Sexton 1988 [25], where the use of semen filtration had a positive effect on the motility of native semen, and in Rad et al. 2016 [26], who reported a decrease in the motility of rooster spermatozoa during short-term storage. The same trend was noted in the results of sperm chromatin integrity assessment; the most favorable results were obtained in the filtered semen group ($p < 0.01$).

The results obtained by assessing the degree of aggregation of spermatozoa proved the advantage of the filtration method, as more "physiological" for the cell. The number of spermatozoa in conglomerates (agglutinated) in the semen with filtration was not more than 5.0% of the total number of cells, against 27.0% in the LCM control sample; in the semen with centrifugation it reached 15.0%.

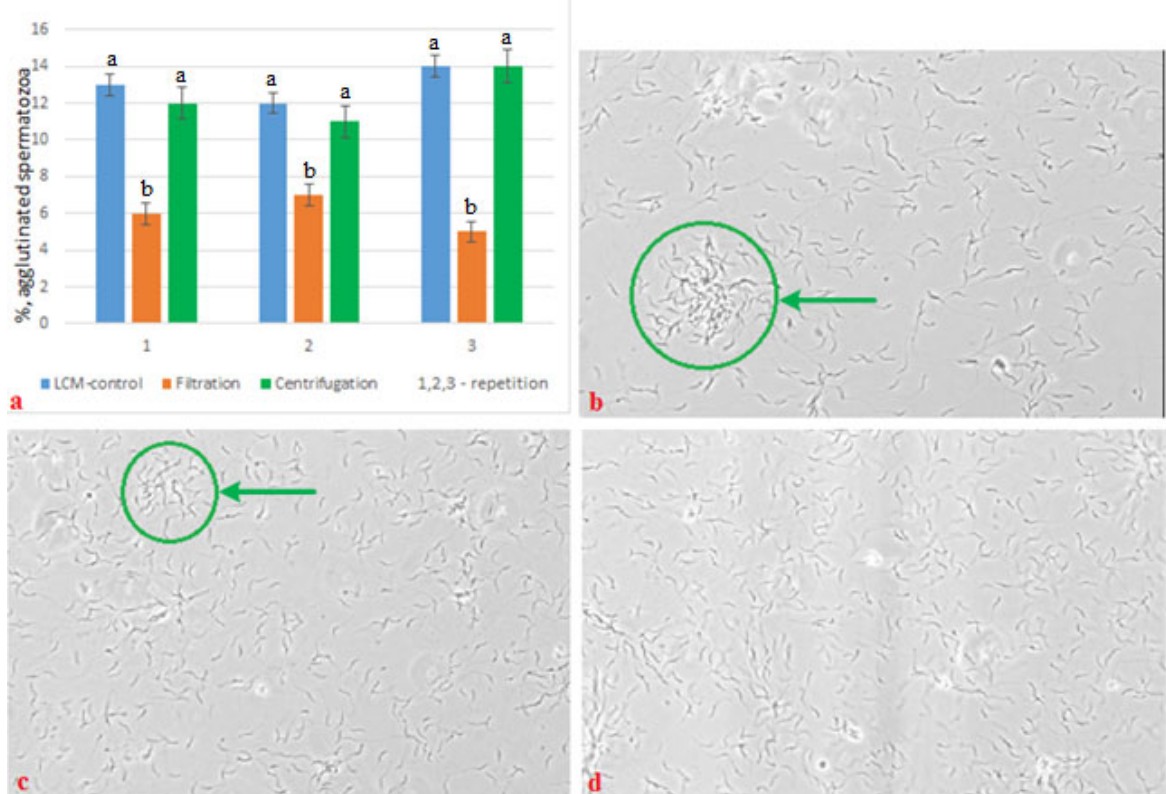

**Figure 4.** The degree of agglutination of frozen/thawed semen, depending on the method of its processing: (**a**) the degree of agglutination (%) of frozen/thawed semen, depending on the technological factors of preparation; (**b**) size of conglomerates of agglutinated frozen/thawed semen from aliquot I (LCM control); (**c**) the size of the conglomerates of the agglutinated frozen/thawed semen from aliquot II (filtration); (**d**) the size of the conglomerates of the agglutinated frozen/thawed semen from aliquot III (centrifugation). (←← designation of the area of agglutinated spermatozoa) ($p < 0.05$).

One of the most important criteria for quality rooster semen is its microbiological purity. Microbiological analysis of semen is not a mandatory step in the routine protocol for artificial insemination of hens [27], which may require revision of the conditions of modern poultry farms. Bacteriospermia has been correlated in many studies with a significant increase in sperm DNA fragmentation [28,29]. The size of bacteria characteristic of semen contamination and the pore size of the filter membrane (0.2 μm) used in the experiment were comparable for sedimentation of the microorganism on the membrane; it can be assumed that the results of assessing the degree of DNA preservation (in terms of chromatin integrity) of spermatozoa in the semen with the use of filtration during storage for 3 h were 1.8 times higher due to a decrease in the total contamination of the filtered semen, since bacterial contamination of the semen causes excessive accumulation of metabolic products in it during storage. Thus, an excessive level of ROS triggers cascade oxidative reactions in the spermatozoon and leads to its damage (DNA fragmentation).

**Experiment 2.** The composition of the cytosol of the native semen after equilibration with the diluent LCM control 1:1 and during short-term storage.

Carbohydrates and polyols play a critical functional role in maintaining the energy reserves and structural integrity of the spermatozoon. In this regard, the dynamics of their content in the cytosol of spermatozoa were studied depending on the technological stages.

In aliquot I (native undiluted semen), the total number of carbohydrates and polyols of the cytosol was 0.04 mg/mL, in aliquot II (after equilibration with the LCM control diluent)—0.096 mg/mL, in aliquot III (after storage for 3 h with the LCM control diluent)—0.024 mg/mL, and the ratios of carbohydrates varied significantly (Figure 5). Inositol was the main component in the composition of the cytosol of the native spermatozoa —75.6% of the total content of the carbohydrate and polyol component of the cytosol. The proportion of fructose in the native semen before dilution was 1.2%, and after equilibration with the diluent it increased to 29.8%.

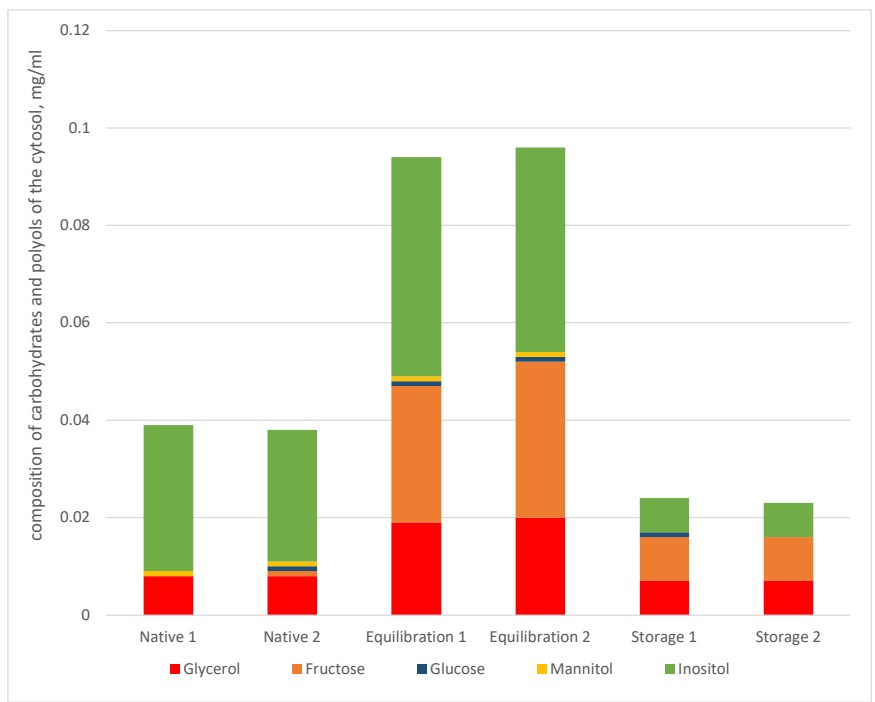

**Figure 5.** Carbohydrate and polyol composition of the cytosol of rooster spermatozoa depending on technological factors (native semen, equilibration, storage for 3 h) in duplicate.

The greatest changes in composition were noted for the following carbohydrates at the stage of short-term storage for 3 h: the content of inositol decreased during storage by 6.5 times (from 0.030 mg/mL to 0.007 mg/mL) and the content of fructose increased by 3.5 times (from 0.001 mg/mL to 0.009 mg/mL) compared to native undiluted semen.

Inositol is an important natural compound, the action of which as an antioxidant is presented by numerous studies [30]. In studies conducted by Condorelli et al. on human cells [31–33], a positive effect of myo-inositol on the potential of the mitochondrial membrane of spermatozoa was shown, which confirmed the significance of its level in the cytosol of the spermatozoon. Being a fundamental element of membranes, inositol is also involved in protein osmoregulation and phosphorylation, and is involved in transduction mechanisms that control calcium levels in the cytoplasm [30].

The negative impact of oxidative stress on semen can be reduced by introducing external natural antioxidants in the diet of males [34,35] or by introducing them into the composition of synthetic cryoprotective media for the semen [15,36]. The present study shows the role of intracellular antioxidants that contribute to the preservation of cellular integrity and functionality during storage as well as freezing/thawing.

## 4. Conclusions

The technological stages of semen preparation for short-term and long-term storage have a significant influence on its quality. The obtained results prove that filtration of rooster semen can be an effective additional step in the preparation of semen for artificial insemination and/or for short-term storage, as well as for long-term storage under conditions of liquid nitrogen. The stage of semen filtration proposed in the study contributes to mechanical purification, that is, the elimination of factors for the occurrence of agglutination centers, increase in oxidative reactions, and the growth of microbial flora. The sizes of common pathological forms of bacteria, such as *Streptococcus* spp. and *Escherichia coli*, are 0.4–1.0 μm and 0.4–08 μm, respectively, which exceeds the pore size of the filter membrane used when filtering the semen in the experiment (0.2 μm).

For the purpose of a deeper study of the processes occurring at the cellular level during storage, the carbohydrates and polyol content in the cytosol of rooster spermatozoa was determined for the first time for the "*native semen-equilibrium-short-term storage*" cycle. The greatest change in the carbohydrate-polyol composition of the cytosol was found for inositol—4.3 times. Such a decrease in the content of inositol in the cytosol during 3 h of storage is a reflection of its importance in the cell as a possible antioxidant, a regulator of osmotic balance, and in a number of intracellular signaling cascades. Based on the results of the chromatographic analysis of the composition of soluble carbohydrates and polyols of the spermatozoa cytosol, a conclusion was made about the predominant role of inositol as the main antioxidant in the cytosol of spermatozoa, which makes it biologically justified to introduce inositol into the composition of synthetic diluents, including cryoprotective ones.

These results form the basis for a more detailed study of the composition of avian spermatozoa cytosol in order to improve the technology of cryogenic storage of genetic resources in vitro.

**Author Contributions:** Conceptualization and methodology, Y.S.; validation, Y.S. and E.F.; formal analysis, Y.S.; investigation, Y.S; resources and data curation, Y.S.; writing—original draft preparation, Y.S.; writing—review and editing, O.S. and E.F.; visualization, Y.S.; supervision, O.S.; project administration, O.S.; funding acquisition, O.S. All authors have read and agreed to the published version of the manuscript.

**Funding:** This research was funded by the Ministry of Science and Higher Education of the Russian Federation, topic №121052600357-8.

**Institutional Review Board Statement:** This study was conducted according to the guidelines of the Declaration of Helsinki and approved by the Institutional Review Board (or Ethics Committee) of the Russian Research Institute of Farm Animal Genetics and Breeding—Branch of the L.K. Ernst Federal Research Center for Animal Husbandry (RRIFAGB).

**Informed Consent Statement:** Not applicable.

**Acknowledgments:** We thank the Centre for Collective Usage "Genetic collection of rare and endangered chicken breeds"and the Russian Research Institute of Farm Animal Genetics and Breeding—Branch of the L.K. Ernst Federal Research Center for Animal Husbandry (RRIFAGB).

**Conflicts of Interest:** The authors declare no conflict of interest.

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
