# Peer review of "Influence of Technological Stages of Preparation of Rooster Semen for Short-Term and Long-Term Storage on Its Quality Characteristics"

_cimb, doi:10.3390/cimb44110374_

Round 1

Reviewer 1 Report

The paper presented the influence of technological stages of preparation of rooster semen for short-term and long-term storage on its quality characteristics. The subject is interesting but, there are minor corrections for acceptance and approval.

1. Correct the grammatical errors, for example, in lines 18, 19, 34, 36, 43, 44, 63, 335, 340, 347, 350, 360..

2. The objective of the study should be stated at the end of the introduction as a paragraph (not as separate items), lines 86 to 93.

3. In Fig. 4, graph (a) of significance and non-significance levels should be determined.

4. To promote the discussion from other related references including...

Superior effect of broccoli methanolic extract on control of oxidative damage of sperm cryopreservation and reproductive performance in rats: A comparison with vitamin C and E antioxidant

The effects of broccoli and caraway extract on serum oxidative markers, testicular structure and function, and sperm quality before and after sperm cryopreservation

5. The conclusion should be more concise and practical.

6. Identify the limitations of the research.

7. Provide additional suggestions for future studies.

Author Response

Dear reviewer!

Thank you for such an attentive reading of our article and for the favorable feedback on the results of our work. We express our deep gratitude to you for your comments and below are our detailed answers.

  1. Correct the grammatical errors, for example, in lines 18, 19, 34, 36, 43, 44, 63, 335, 340, 347, 350, 360..

Comments taken into account, corrections made.

  1. The objective of the study should be stated at the end of the introduction as a paragraph (not as separate items), lines 86 to 93.

Comments taken into account, corrections made.  

  1. In Fig. 4, graph (a) of significance and non-significance levels should be determined.

Comments taken into account, corrections made. 

  1. To promote the discussion from other related references including...

Superior effect of broccoli methanolic extract on control of oxidative damage of sperm cryopreservation and reproductive performance in rats: A comparison with vitamin C and E antioxidant

The effects of broccoli and caraway extract on serum oxidative markers, testicular structure and function, and sperm quality before and after sperm cryopreservation

Comments taken into account, corrections made.  

  1. The conclusion should be more concise and practical.

If you allow, we will not shorten the "Conclusion" section, from our point of view, this form of presentation more fully reflects the results achieved and their significance. 

  1. Identify the limitations of the research.

Comments taken into account, corrections made.  

  1. Provide additional suggestions for future studies.

Our proposal to continue research in lines 366-370.

Sincerely,

The team of authors.

Reviewer 2 Report

The title of the manuscript should be changed like this: Influence of rooster semen collection methods for preparation for short-term and long-term storage on its quality characteristics.

In the section on the design of technological stages of experiments 1 & 2, the authors should clarify the number of ejaculates that were pooled before the division into 3 aliquots. (page 3, lines 113 and 122). Also, in the same section, the threshold of the selected ejaculates in terms of the percentage of total motility and concentration should be mentioned.

In the section on the assessment of semen quality in the field, the authors should clarify the scale of the assessment of total motility. (page 3, line 135).

In the section on the viability of spermatozoa of native and frozen/thawed semen, the authors should describe the CASA settings for the evaluation of spermatozoa motility. Also, it is interesting to know why they do not use all the kinematic parameters that CASA system measures {i.e: Path Velocity-VAP, Straight Line Velocity-VSL, Curvilinear Velocity-VCL, (Straightness-STR) and (Linearity -LIN)}. (page 4, line179).

In the section Results and Discussion, Experiment 1, Table 1., the authors should do a better presentation of the data. They could write mean±SD in every column and n=22 ejaculates. In table 2 the variability of semen quality indicators as CV could be shown in a column.

Page 9, line 280, write: The results

Author Response

Dear reviewer!

Thank you for such an attentive reading of our article and for the favorable feedback on the results of our work. We express our deep gratitude to you for your comments and below are our detailed answers.

  • The title of the manuscript should be changed like this: Influence of rooster semen collection methods for preparation for short-term and long-term storage on its quality characteristics.

Thank you for the proposed version of the title of the article, but in our opinion it does not reflect its content. We studied not "rooster semen collection methods" but "technological stages of preparation".

  • In the section on the design of technological stages of experiments 1 & 2, the authors should clarify the number of ejaculates that were pooled before the division into 3 aliquots. (page 3, lines 113 and 122).

Comments taken into account, corrections made.

  • Also, in the same section, the threshold of the selected ejaculates in terms of the percentage of total motility and concentration should be mentioned.

Comments taken into account, corrections made. 

  • In the section on the assessment of semen quality in the field, the authors should clarify the scale of the assessment of total motility. (page 3, line 135).

Comments taken into account, corrections made. 

  • In the section on the viability of spermatozoa of native and frozen/thawed semen, the authors should describe the CASA settings for the evaluation of spermatozoa motility. Also, it is interesting to know why they do not use all the kinematic parameters that CASA system measures {i.e: Path Velocity-VAP, Straight Line Velocity-VSL, Curvilinear Velocity-VCL, (Straightness-STR) and (Linearity -LIN)}. (page 4, line179).

The kinematic parameters proposed by you for assessing the movement of rooster spermatozoa, as our experience shows, are not so significant parameters due to the biological species characteristics of chickens and they are not related to the motor activity of spermatozoa in the chicken genital tract and their fertilizing ability. We found confirmation of our point of view in many works (for example, https://doi.org/10.1016/j.psj.2020.10.076), so we did not use these parameters in our study.

  • In the section Results and Discussion, Experiment 1, Table 1., the authors should do a better presentation of the data. They could write mean±SD in every column and n=22 ejaculates. In table 2 the variability of semen quality indicators as CV could be shown in a column.

Comments taken into account, corrections made.

  • Page 9, line 280, write: The results

Comments taken into account, corrections made.

Sincerely,

The team of authors
